# Study of *Aquilaria crassna* Wood as an Antifungal Additive to Improve the Properties of Natural Rubber as Air-Dried Sheets

**DOI:** 10.3390/polym13234178

**Published:** 2021-11-29

**Authors:** Phattarawadee Nun-Anan, Sunisa Suchat, Narissara Mahathaninwong, Narong Chueangchayaphan, Seppo Karrila, Suphatchakorn Limhengha

**Affiliations:** Faculty of Science and Industrial Technology, Prince of Songkla University, Surat Thani Campus, Surat Thani 84000, Thailand; phattarawadee.Anan@gmail.com (P.N.-A.); sunisa.su@psu.ac.th (S.S.); narissara.s@psu.ac.th (N.M.); narong.c@psu.ac.th (N.C.); seppo.karrila@gmail.com (S.K.)

**Keywords:** *Aquilaria crassna* wood, natural rubber, antifungal, lignin, proteins

## Abstract

Fungal growth on rubber sheets confers inferior properties and an unpleasant odor to raw natural rubber (NR) and products made from it, and it causes environmental concerns. The purpose of the present work was to investigate the effects of *Aquilaria crassna* wood (ACW) on the antifungal, physical and mechanical properties of NR as air-dried sheets (ADS) and ADS filled with ACW. The results show that the ACW-filled ADS had an increased Mooney viscosity, initial plasticity (*P_O_*), and high thermo-oxidation plasticity (i.e., high plasticity retention index PRI). Additionally, superior green strength was observed for the ACW-filled ADS over the ADS without additive because of chemical interactions between lignin and proteins in NR molecules eliciting greater gel formation. A significant inhibition of fungal growth on the NR products during storage over a long period (5 months) was observed for ACW-filled ADS. Thus, it can be concluded that ACW could be applied as an antifungal additive that reduces fungal growth. This is a practically important aspect for the rubber industry, as fungal growth tends to spoil and cause the loss of NR sheets during storage. Moreover, the ACW is active as an incense agent, reducing negative impacts from odors that fungi, on rubber surfaces, release. Therefore, these filled intermediate NR products provide added value through, an environmentally friendly approach, this is pleasant to customers.

## 1. Introduction

Thailand is among the world’s largest producers of natural rubber. Dry natural rubber (NR) is obtained from fresh latex suspension by processing it to block rubber, air-dried sheets, or ribbed smoked sheets. These intermediate rubber products can subsequently be processed into numerous consumer products like gloves, coatings, and tires. NR has excellent flexibility and tensile strength, and a good tear strength, together with low heat buildup [1]. Currently, NR products are rapidly being adopted by the food industry, and are used in contact with food or water, for example in foodstuff conveyor belts that need high flexibility [2]. However, the high moisture content of rubber sheets can enable fungal growth. This is a serious problem in rubber processing that can affect both the processing as well as the eventual properties of raw NR, and further on can also impact the final consumer products.

To overcome such problems, antifungal agents need to be added to the rubber during processing to reduce the growth of fungi, both in raw NR and its vulcanizates. Many studies have assessed effects of alternative natural additives, such as wood vinegar, as coagulating and antifungal agents for the rubber-sheet production process. For instance, the performance of wood vinegar from crude coconut shell as an additive for NR products has been studied [3]. It was found that the composition of wood vinegar (i.e., dominantly acetic acid and phenolic compounds) and its acidity improved the physical properties of raw NR, including plasticity and Mooney viscosity, while also inhibiting the growth of fungi on surfaces of the NR products [3]. Furthermore, using wood vinegar (produced from rubber wood) to inhibit fungi and malodors in NR products was also investigated [4]. The results showed that vinegar from rubber wood reduced the fungal colony counts on rubber sheets and decreased malodors while drying the rubber [4]. Recently, wood vinegars from para rubber wood, bamboo, and coconut shell have been tested as substitutes for a commercial acid (i.e., acetic acid) in coagulating rubber [5]. Regarding the physical and mechanical properties of the NR product, the results showed that the type of wood vinegar coagulant (i.e., para wood, bamboo, or coconut shell vinegar) did not cause significant differences from using the commercial coagulant. In addition, the rubber sheets coagulated with wood vinegar had comparatively less fungal growth than those using acetic acid [5].

*Aquilaria crassna* wood (ACW) is a local natural resource in Thailand, also grown in Vietnam, Laos, and Cambodia for the agar wood market [6]. At present, the fragrant *Aquilaria crassna* is popular in perfumes, decorations, and incenses [7]. However, the ACW residue from perfumery extraction is a waste material that can be considered for use in relatively low-cost rubber products. Moreover, ACW is a lignocellulosic material (i.e., composed mainly of cellulose, hemicellulose, and lignin) with only small quantities of other substances, such as phenols and acetyl groups [6]. Proteins account for approximately 0.5%wt of plant cell walls, where they are crosslinked with lignin [8,9]. It has also been reported that these components in wood contribute to its stiffness, strength and resistance against insects and pathogens (both in the plant and in the eventual wood products) [10]. *Aquilaria crassna* wood has many applications in food, cosmetics, and pharmaceutical products, depending on the purity and the wood source [10,11]. The ACW has been applied in agriculture as a peptizer, an anti-fungal, and an anti-termite substance, improving the longevity of other wood materials; it enables value-added wood products [12]. Therefore, it has potential to serve as an antifungal, also, in raw natural rubber.

In the present work, extracted *Aquilaria crassna* wood (ACW) was used as an environmentally friendly alternative additive for natural rubber in the form of air-dried sheets (ADS). The purpose of the study was to assess the effects of using ACW at various contents (0, 20, 40 and 60 phr) in NR products, under the hypothesis that ACW could enhance the key properties and antifungal characteristics of NR products (or specifically ADS). The ADS with and without ACW was analyzed for dirt, volatiles, nitrogen, and gel content, as well as for plasticity, Mooney viscosity, and plasticity retention index (PRI) together with their tensile properties. Moreover, the antifungal efficiencies of both ADS and ACW-filled ADS were evaluated by recording the area fractions of fungal growth on the rubber surfaces.

## 2. Materials and Methods

### 2.1. Materials

Field natural rubber latex was obtained from the RRIM 600 clonal variety of rubber trees, in a rubber plantation located in Surat Thani province, Thailand. The chemicals for filler dispersion and bentonite were supplied by the BASF Company (Rhineland-Palatinate, Ludwigshafen, Germany). Vultamol, used as dispersant, was produced by S&B minerals GmbH (Kifissia, Athens, Greece). Chemicals for the determination of dirt content, 2-mercaptobenzothiazole and turpentine oil, were produced by Merck KGaA (Darmstadt, Germany) and Vidhyasom CO., Ltd (Phra Nakhon, Bangkok, Thailand), respectively. In addition, boric acid, sulfuric acid (H_2_SO_4_), and potassium sulfate (K_2_SO_4_), used to analyze nitrogen content, were manufactured by Merck KGaA (Darmstadt, Germany). The copper (II) sulfate (CuSO_4_) was obtained from VWR (Fontenay-sous-Bois, France).

### 2.2. Aquilaria crassna Wood (ACW) Powder Preparation

Extracted *Aquilaria crassna* wood was obtained from the perfumery process, in which fragrant substances are extracted. The extracted wood residue was sawn to small pieces and then ground in a ball mill for about 48 h. After that, the powder was passed through a 120-mesh screen before the preparation of an ACW dispersion. The functional groups in the ACW particles were analyzed by Fourier transform infrared spectroscopy (Perkin Elmer Inc., Waltham, MA, USA).

### 2.3. Preparation of Aquilaria crassna Wood Dispersion

*Aquilaria crassna* wood, as a filler, was first mixed with distilled water using a dispersing agent (Vultamol). The filler dispersion was prepared by ball milling, using ceramic balls for 72 h, to ensure the breakdown of filler aggregates (Table 1).

### 2.4. Air-Dried Sheet Preparation

Field latex was tapped and collected from RRIM 600 clones in Surat Thani, Thailand. The latex was first diluted with clean water to 20% dry rubber content (%DRC). Then, the diluted latex was coagulated by using 5% *w*/*v* formic acid and left for 3 h to solidify. After that, the soft coagulum formed was passed through a nip between two steel rolls and washed with clean water before drying in a hot air oven at 50 °C until a constant weight was reached. Finally, air-dried sheets (ADS) were obtained.

### 2.5. Preparation of Aquilaria crassna Wood-Filled Rubber Sheet

The rubber latex of 20% DRC was prepared by diluting the initial field natural rubber latex with about 35% DRC using clean water. Then, 30% ACW dispersion was added into the latex to achieve the targeted contents (0, 20, 40 and 60 phr) in an aqueous solution/dispersion, and the mix was stirred well using a mechanical stirrer (Onilab, San Francisco, CA, USA) at 50 rpm for 10 min to homogenize the dispersion. The mixture was then added with 10% *w*/*v* formic acid to coagulate it in the coagulation tank and was allowed to solidify for 3 h. The coagulum was pressed by rollers to make a rubber sheet, which was further dried at 50 °C in a hot-air oven until constant weight. Then, *Aquilaria crassna* wood (ACW)-filled dry rubber sheets were obtained.

### 2.6. Preparation of NR Thin Films

NR thin films were prepared from NR sheet with a compression-molding machine (PR1D-W400L450 PM, Chareon Tut Co., Ltd., Bang Phli, Samut Prakarn, Thailand). First, the NR sheets were compressed at 160 °C for 10 min. After that, the films were immediately cooled down to room temperature, under pressure for 10 min by a cooling system. The rubber films were left at room temperature for one week before use, to allow the residual strains from molding to relax.

### 2.7. Characterization

#### 2.7.1. Fourier Transform Infrared Spectroscopy (FTIR)

FTIR spectra of the rubber sheets were recorded with a Perkin-Elmer Spectrum spectrometer (Perkin Elmer Inc., Waltham, MA, USA). The analysis was carried out over the wavelength range of 4000–400 cm^−1^ with 32 scans acquired per recorded spectrum, at a resolution of 4 cm^−1^.

#### 2.7.2. Analysis of ACW

The chemical composition of ACW powder was characterized by X-ray fluorescence spectrometry (XRF) (PW2400, Philips & Co., Eindhoven, North Brabant, The Netherlands). First, ACW powder was dried to a constant weight in a hot-air oven at 100 °C for 14 h. After that, the sample was treated in a muffle furnace by heating from 50 °C to 1000 °C. It was then mixed with a binder (WAX C) at a weight ratio of 2:1 and further compressed to a thin sheet. Finally, the sample sheets were characterized by XRF.

#### 2.7.3. Characterization of Air-Dried Sheet (ADS)

The dirt content of air-dried sheets (ADS) was measured according to ISO 247-2, as described elsewhere [13]. The homogenized rubber sample (about 10 g) was first cut into small pieces and then immersed in a volumetric flask containing 200 mL of a mixture of turpentine oil (boiling point 154–198 °C) and 0.5 g of 2-mercaptobenzothiazole, as a peptizer. After that, the solution was left at room temperature for 48 h and then heated at 130 ± 5 °C to complete dissolution. The obtained solution was then separated through 45-mm sieves, after which the dirt retained was washed, dried and its weight was recorded. Finally, the weight percent of dirt content was calculated as follows:(1)$$\mathrm{Dirt}\;\mathrm{content}\;(\%\;\mathrm{wt})=\frac{B-A}{W}\times 100$$
where *A* is the mass of the sieve (g), *B* is the mass of the sieve containing dirt (g), and *W* is mass of the rubber sample (g).

In addition, determination of the ash contents of the ADS samples was performed according to ISO 247-2, as described elsewhere [14]. First, the rubber sample was weighed and placed inside a crucible, and this was placed in a heat-treatment oven, an Optic Ivymen System (SNOL 3/1100 LHM01, Utena, Lithuania), at 550 °C, in order to achieve total oxidation of the sample. After that, the remaining ashes corresponded to the inorganic components in the ADS sample. The ash content was calculated as follows:(2)$$\mathrm{Ash}\;\mathrm{content}\;(\%\;\mathrm{wt})=\frac{m_{2}}{m_{1}}\times 100$$
where *m*_1_ and *m*_2_ are the initial mass of rubber sample (g) and the final mass (g) of rubber sample after the oven treatment (ash), respectively.

Moreover, the volatile matter content (VM) of the ADS was also investigated according to ISO 248, as described elsewhere [15]. About 10 g of homogenized rubber was passed through a two-roll mill to control the thickness at less than 2 mm. The sample pieces were placed in a conventional circulating-air oven at 100 ± 5 °C for 4 h. After that, the rubber sample was cooled in a desiccator at room temperature for 30 min before weighing the rubber sample, and the volatile matter content was calculated as follows:(3)$$\mathrm{Volatile}\;\mathrm{matter}\;\mathrm{content}\;(\%\;\mathrm{wt})=\left(\frac{W_{1}-W_{2}}{W_{1}}\right)\times 100$$
where *W*_1_ is the initial mass of the sample and *W*_2_ is the mass of the sample after heating.

#### 2.7.4. Gel Content

The gel content in ADS was investigated in accordance with ISO 1166, as described elsewhere [16]. First, an ADS sample of about 0.1 g was cut into small pieces and suspended in 30 mL of toluene, followed by shaking for a few minutes, and was left at 25 °C for 20 h without stirring. After that, the insoluble rubber was separated by centrifuging at 22,000 *g* and the liquid fraction was removed from the tube. The remaining gel fraction was washed with acetone and then dried at 110 °C for 1 h. Then, the gel content of NR was calculated as follows:(4)$$\mathrm{Gel}\;\mathrm{content}\;(\%\mathrm{wt})=\left(\frac{m_{0}-m_{1}}{m_{0}}\right)*100$$
where *m*_1_ and *m*_0_ are the dry weights after extraction and before extraction, respectively.

#### 2.7.5. Measurement of Initial Plasticity

The plasticity of ADS was determined using a Plastimeter H-01 (CG Engineering Ltd., Part, Sam Khok, Pathum Thani, Thailand), in accordance with ISO 2007. First, the ADS was masticated and homogenized on a two-roll mill (YFCR 6″, Yong Fong machinery CO., LTD, Mueang Samut Sakhon, Samut Sakorn, Thailand) at ambient temperature. The sample was sheeted-out to 3.2–3.8 mm of thickness. After that, test pieces were cut from the thin sheet with a specimen-cutting press. Two sample sets were prepared for the determination of initial plasticity (*P_O_*) and the plasticity retention index (PRI), using three replicates in each test.

#### 2.7.6. Plasticity Retention Index (PRI) Determination

PRI is a measure of resistance of the NR product to thermal oxidation. The rubber pieces were inserted into an ageing chamber for 30 min at 140 °C, removed, and allowed to cool at room temperature. After that, the test pieces were immediately compressed by a plastimeter under a constant compressive force of 100 N at 100 °C for 15 s. The plasticity of the not-aged sample (*P_O_*) and plasticity of the aged sample (*P*_30_) were obtained by measuring their thickness changes. Finally, the median of three replicate samples was taken as the plasticity value. The plasticity retention index (PRI) was calculated as follows:(5)$$\mathrm{PRI}=\left(\frac{P_{O}}{P_{30}}\right)\times 100$$
where *P_O_* is the plasticity before aging in the oven and *P*_30_ is the plasticity after aging in the oven.

#### 2.7.7. Mooney Viscosity

Mooney viscosity indicates the processability of an elastomer and it was measured by using a MV-2020 Mooney viscometer (Montech, Shinjuku, Tokyo, Japan) with a large rotor size, at 100 °C, and according to ISO 289.

#### 2.7.8. Tensile Properties

The thin NR sheets were cut to dumbbell shapes with die type 5A, in accordance to ISO 527. Then, their tensile properties were measured using a tensile testing machine (Zwick Roell, Ulm, Germany) at room temperature with a crosshead speed of 500 mm/min and a 100-N load cell. The measurement was repeated five times for each type of rubber sample.

#### 2.7.9. Antifungal Performance

The antifungal performance of the rubber sheets was investigated by measuring fungal growth area on the pre-dried rubber sheet surfaces after leaving them for 5 months. Area percentage of the fungal growth was calculated, as described elsewhere [17], by:(6)$$\%\;\mathrm{Fungal}\;\mathrm{growth}\;\mathrm{area}=\frac{A_{fungi}}{A_{NR}}\times 100$$
where *A _fungi_* is the fungal growth area on the rubber sheet surface (cm^2^) and *A_NR_* is the total area of rubber sheet surface (cm^2^).

## 3. Results and Discussion

### 3.1. Characterization of the ACW Powder

The functional groups present in the *Aquilaria crassna* wood (ACW) powder are shown in Figure 1. It is clearly seen that ACW showed a broad peak at the wave number 3340 cm^−1^, assigned to –OH stretching vibrations from the phenol, alcohol, and acid groups present in lignin of ACW (Table 2) [18,19]. These also caused the ACW absorption bands at 1730, 1620, and 1029 cm^−1^, seen in Figure 1 and Table 2. The absorption peak at the wavelength range of 1740–1710 cm^−1^ is attributed to the carbonyl group (C–O) stretching vibrations in the esters and acids [19,20]. In addition, the broad peak around 1620 cm^−1^ indicates aromatic rings. Furthermore, the absorption peak at 1029 cm^−1^ is from the stretching vibrations of C–O [19].

The chemical composition of *Aquilaria crassna* wood was analyzed by X-ray fluorescence spectrometer (XRF), with the results shown in Table 3. It is seen that *Aquilaria crassna* had CHNO content of about 98%. These major components correlated well to lignin’s structure and a small quantity of proteins present in cell walls of plants (or in ACW) [10,19], which may affect in the properties of ADS products.

### 3.2. FT–IR Spectra

Figure 1 and Table 2 show the FT–IR spectra for ADSs with various ACW contents (0, 20, 40, and 60 phr). It can be seen that the typical absorption peaks of natural rubber appeared at 1663 and 832 cm^−1^, attributed to C=C stretching vibrations and =CH out-of-plane bending of cis-1,4-isoprene units in the natural rubber, respectively [21,22]. Reductions and shifts of the broad, strong –OH stretching of alcohol, phenol, and acid in lignin, at about 3100–3600 cm^−1^, was observed [19]. The decrease of the band at 3100–3600 cm^−1^ can be attributed to intramolecular hydrogen bonding between lignin structures [23]. Furthermore, the overlapped absorption peaks at 1630 and 1540 cm^−1^ could possibly be due to proteins present in ACW [9,10], because the intensities of these bands increased with ACW content. However, the absorption peaks at 3280, 1630,and 1540 cm^−1^ were partly affected by mono- and di-peptides of proteins present in *Hevea* NR [24,25]. This relates to the elemental composition in ACW, as given in Table 3, attributed to the –NH group in the amines and amide II (N–H stretching vibrations) of proteins, respectively [24]. The peak ratios between –NH group (3280+1540 cm^−1^) and the symmetric CH_2_ stretching vibrations (2963 cm^−1^) were used to estimate the nitrogen content in the NR samples, with the results shown in Table 4. It is clear that the ACW60 showed the highest peak ratio for proteins in NR, while the ACW40, ACW20 and ACW0 gave lower intensity ratios for proteins in the NR sample. It is notable that nitrogen content was directly used to estimate the level of proteins in raw NR products [25]. This implies that ADS with varying ACW content contains detectable proteins (i.e., protein in ACW) even though these are of different types than those present in *Hevea* NR. This result matches well the percentage of N atoms in *Aquilaria crassna* wood, as seen in Table 3. Also, it matches well the level of protein content calculated from the percentage of N atoms, as analyzed by the Kjeldahl method (Figure 2). Thus, it is interesting that it is possibly the amino acids of proteins in ACW and proteins in *Hevea* NR that react in ways that impact the various properties of raw NR.

Moreover, increasing *Aquilaria crassna* content caused changes in the absorbance ratio 1240 cm^−1^/2963 cm^−1^ (where the band at 1240 cm^−1^ is assigned to C–O stretching in the phenolic compounds of lignin, and the band at 2963 cm^−1^ is assigned to symmetric CH_2_ stretching vibrations) [19,26]. It is seen that the ratio 0.25 for ACW60 case is higher than those of the other samples, while the ratio is 0.15 for ACW0 (without any *Aquilaria crassna* in the ADS) (Figure 1). This intensity ratio could be attributed to crosslinking between ACW and protein molecules in NR, or to the filler–filler interactions of ACW. These possible interactions are reflected by an increase in hydrogen bonding in the filled NR [27], which may improve the physical properties of rubber sheets.

### 3.3. Physical Properties of Raw NR

The effects of ACW content on physical properties, including dirt, ash and volatile matter (VM) contents in ADS, are summarized in Table 4. The dirt content of ADS is a measure of the contamination level of the rubber sheets during the processing of latex to dry NR. It is seen that the dirt content of the rubber sheets increased with ACW content. The ACW60 sample showed the highest dirt content, of about 8.23 %wt, followed by the ACW40 (6.97 %wt) and ACW20 samples (6.88 %wt), in rank order (Table 4). In addition, ACW0 had the lowest dirt content, of about 0.03 %wt (Table 4). This is because it is a major component in ACW, as an additive in NR, and may have caused increased ash content in the NR sheets [10,28]. Therefore, ACW was the main contributor of the ash content in the filled rubber sheets. Furthermore, in most cases, a high ash content in NR is directly correlated with a high dirt content, as was also seen in the present work. This relation between dirt and ash contents in rubber sheets is shown in Figure 3. As is clearly seen, both the dirt and ash contents of the ADS increased with ACW content, due to *Aquilaria crassna’s* composition and some soluble impurities present in the *Hevea* NR [10,29].

Table 4 and Figure 3c show the volatile matter (VM) contents in rubber sheets with different ACW contents. The volatile content of dry rubber is related to its moisture, as well as to its contamination with volatile matter during rubber processing (i.e., during coagulation of rubber latex) and showed the same trend as ash and dirt contents, as seen in Figure 3. The VM content increased with ACW content. The VM in ACW60 sample was about 5.08 %wt, which was higher than in the other samples. This behavior of the ADS product was caused by hydroxyl groups (–OH), as water absorption sites [30]. This may be associated with antifungal activity on the NR surface; due to their high moisture content, NR products are susceptible to fungal growth [17].

### 3.4. Plasticity of Rubber

Table 5 summarizes the initial plasticities (*P_O_*) and Mooney viscosities of the ADSs. *P_O_* and Mooney viscosity represent the ability of rubber to deform [3]. It was observed that the ADS without ACW (or ACW0) gave the lowest *P_O_*, of about 40 (Table 5). On the other hand, *P_O_* increased with ACW content. The ACW60 sample gave the highest *P_O_* of about 61, followed by ACW40 and ACW20, at about 50 and 40, in rank order (Table 5). Increasing ACW content increased *P_O_* in the NR products, due to phenolic compounds in ACW that can interact with the amine groups of NR molecules [31].

Furthermore, Mooney viscosity increased with ACW content in the rubber sheets, showing a trend similar to that for initial plasticity, as shown in Figure 4. The Mooney viscosity had the rank order ACW60 > ACW40 > ACW20 > ACW0. When the ACW content was 60 phr, the Mooney viscosity was the largest (96, Table 5). The increases in both *P_O_* and Mooney viscosity were caused by (I) the intramolecular hydrogen bonding of lignin, and (II) the covalent and hydrogen bonds between lignin and protein in ACW [32,33]. Possibly, there were non-covalent bonds between NR and lignin [34] and chemical interactions between NR proteins via the active functional groups (or –OH groups) of lignin in ACW, as illustrated in Figure 5. These results match well the gel contents observed in ADS, which given in Table 5. It is notable that the gel content in NR represents the interactions between proteins and other active functional groups via hydrogen or other bonding [35,36]. In Table 5, it can be observed that the ACW60 gave the highest gel content, of about 50%wt followed by the ACW40, ACW20, and ACW0 samples (31, 20, and 8 %wt). This confirms the increased intramolecular hydrogen bonding of lignin in ACW with ACW content, by the interactions of lignin with proteins in rubber that contributed to the gel observed in NR. Furthermore, this result matches well the small shift in wavelengths at 1443–1445 cm^−1^ for WAC20, ACW40, and ACW60, as compared to the not-filled case (ACW0), associated with the vibrations of rubber molecules [34]. This shift indicates non-covalent interactions between the rubber molecules and lignin, and adsorption of NR onto the lignin in ACW [34] that increased with ACW content.

The oxidation resistance of raw NR is indicated by the plasticity retention index (PRI), summarized in Table 5. If the value is high, it shows that the rubber resisted thermo-oxidation. The results show that increasing ACW content improved the thermo-oxidation resistance of natural rubber. ACW60 had the highest PRI at about 110, indicating the best oxidation resistance, while ACW40 and ACW20 also showed higher PRI values (about 95 and 101) than that of the rubber sheets without ACW content (90). The highest PRI was found for the rubber sheet with the most ACW, so these rubber sheets resisted thermal oxidation comparatively well. The PRI for rubber sheets improved with ACW content, possibly, because of phenolic compounds and proteins in ACW [3,37].

### 3.5. Tensile Properties

Figure 6 shows the stress–strain behavior of the NR films. It was observed that the ACW0 sample (without ACW) gave the lowest green strength, of about 0.27 MPa, while it exhibited an excellent elongation at break of approximately 669%. Furthermore, it should be noted that increased ACW content enhanced the green strength of ADS and conferred reduced elongation at break relative to the control cases without ACW. ACW60 showed the highest green strength, of about 2.19 MPa, while the ACW40 and ACW20 samples had lower green strengths (1.84 and 1.29 MPa, respectively, Table 6) than ACW60 (Figure 6). This is due to a high gel content (51%wt, Table 5), which encouraged the strength and stiffness of NR product [38]. On the other hand, the elongation at break of the rubber sheets decreased with ACW content (Figure 6). The ACW60 sample showed the lowest elongation at break (about 158%, Table 6) followed by the ACW40, ACW20, and ACW0 samples. This may be attributed to ACW, as a filler, inhibiting the orientation-induced crystallization of NR molecules [39], which decreased the elongation at break of the ADS.

Table 6 summarizes the 100% and 300% moduli and toughnesses of the NR sheets as a function of ACW content (i.e., 0, 20, 40, and 60 phr). It is observed that ACW0 showed the lowest 100% and 300% moduli, due to its having the lowest gel content (or low intramolecular, physical, and chemical interactions) of about 8%wt (Table 5). Apparently, high lignin and protein in ACW60 acted as reinforcing fillers and enhanced the rigidity of the filled ADS. Possibly, the physical and chemical interactions between lignin and proteins in NR produced strong natural crosslink formations with high 100% and 300% moduli, along with greater stiffness, as shown in Figure 7. Therefore, it can be concluded that the use of ACW as an environmentally friendly filler increased the 100% and 300% moduli and the green strengths of the filled ADS products.

### 3.6. Antifungal Performance of the Rubber Sheet

The antifungal performance of the ADS, as a function of ACW content, was measured by the area fractions of fungal growth on the ADS, when the rubber sheets were left at ambient conditions for 5 months. The percentage of fungal growth area was calculated according to Equation (5). It was observed that the dry rubber sheet had a yellow–brown color and was without fungi (white spots on the NR surface) before the storage for 5 months (Figure 8a). After storage for 5 months, the color of the rubber sheet had changed to black-and-brown and showed some white spots (i.e., fungi or bacteria) on the surfaces. The fungi grew well on the ACW0 sample, as shown in Figure 8b. In contrast, the rubber sheets with ACW at 20-phr loading had quite similar appearances (i.e., surface and color) before and after storage for 5 months. This suggests that ACW could inhibit fungal growth on surfaces of rubber sheets, as shown in Figure 8d, having high antifungal activity. However, the fungal growth area fractions were 65, 69, and 73% for ACW60, ACW40, and ACW20, respectively, while the highest fungal growth area fraction was 97%, for the ACW0 sample (without *Aquilaria crassna*, see Table 7). This indicates that ACW filler in the ADS had antifungal activity in the rubber product due to its composition, including, especially, phenolic compounds [4], which are antimicrobial [40]. Moreover, it had a strong antifungal effect on microorganisms [41].

The growth of fungi on the rubber sheet surfaces is shown in Figure 9. It can be seen that the fungi were well distributed on ACW0 (without ACW filler) in Figure 9a. In contrast, the rubber sheets with ACW at 20, 40, and 60 phr did not have growth of fungi on their sheets’ surfaces, as shown in Figure 9b–d, respectively. This confirms that the phenolic compounds in ACW inhibited fungal growth on the dry rubber products [14]. Using ACW as a filler in rubber sheets could improve their antifungal performance.

In addition, the changes in initial plasticity (*P_O_*) and Mooney viscosity (Figure 10) of all ADS samples were tested after storage for 5 months (Figure 10a,b). The results correlated well with the ACW content, the quantity of proteins (Figure 2), and gel content (Table 5). Thus, it is concluded that *Aquilaria crassna* wood is suitable for use as a filler, having not only contributed to improving the plasticities and viscosities of the NR products by the provided lignin and proteins, but also in having improved the antifungal performance of the filled ADS product.

As is well known, in block rubber processing, unpleasant odors are released during the storage of fresh or dried cup-lump rubber, and from thermal degradation during the drying of shredded rubber. These odors from rubber factories have mainly been attributed to noxious volatile components, which are discharged into the atmosphere through a chimney during the drying stage in block rubber processing [42]. The odors have often led to complaints from the public, who are also irritated by other environmental problems associated with the rubber industry. Therefore, ACW powder, as an additive against fungi, could also serve as an incense, providing a pleasant scent to the NR product, especially in the case of cup lump, which will be studied in further work.

## 4. Conclusions

The present work studied the effects of *Aquilaria crassna* wood (ACW) as an alternative antifungal additive on the properties of a raw NR product (i.e., air-dried sheets, ADS). Various properties of raw NR in ADS, including initial plasticity (*P_O_*), plasticity retention index (PRI), Mooney viscosity and tensile properties (i.e., 100% and 300% moduli and green strength) were also considered. The results showed that ACW-filled ADS had improved *P_O_*, Mooney viscosity, and tensile moduli, as well as green strength, because of the hydrogen bonding of lignin in ACW with proteins in NR. Improved resistance to thermo-oxidation and better retention of plasticity were also observed for the filled ADSs, because of phenolic compounds in the ACW. Moreover, ACW’s antifungal activity inhibited or retarded fungal growth on the rubber sheets, as compared with the control without ACW, during storage for a long period (5 months), in a dose-dependent manner. Thus, ACW is an alternative antifungal additive, improving ADS properties and preventing fungal growth on NR products, while also acting as an incense to provide a pleasant scent. These characteristics represent important advantages to the rubber industry in its relationships with neighbors and customers. Additionally, rubber tree plantations and end users of the NR products stand to benefits therefrom.

## Figures and Tables

**Figure 1 polymers-13-04178-f001:**
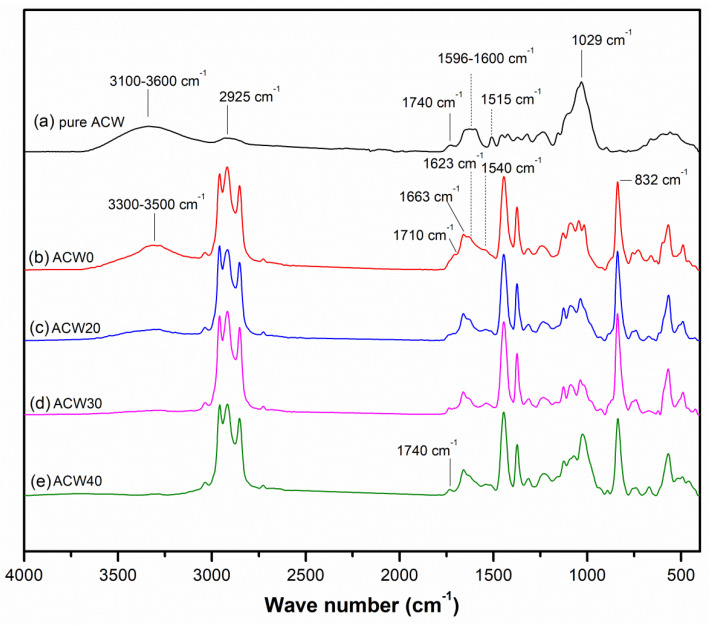
FTIR spectra of pure *Aquilaria crassna* wood (ACW) (**a**), and for air-dried sheets with various ACW contents: 0 phr (**b**), 20 phr (**c**), 40 phr (**d**), and 60 phr (**e**).

**Figure 2 polymers-13-04178-f002:**
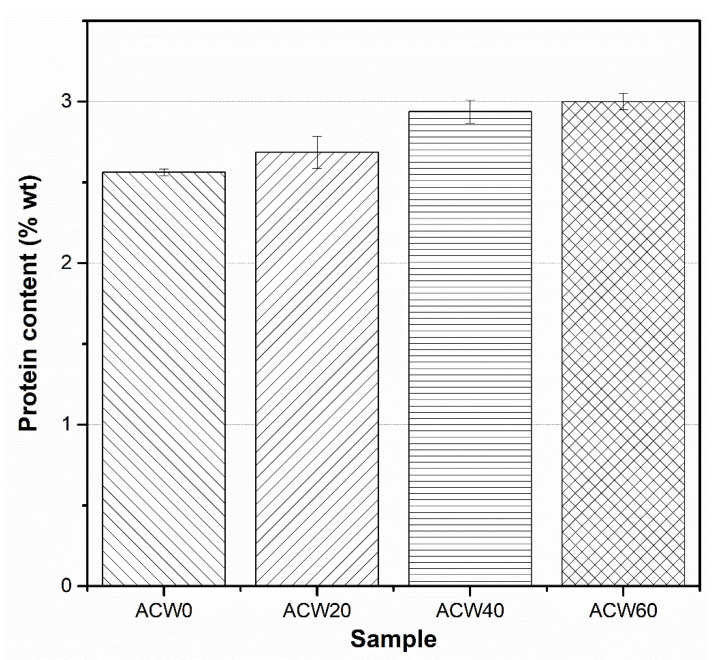
Protein content of NR as a function of *Aquilaria crassna* wood content.

**Figure 3 polymers-13-04178-f003:**
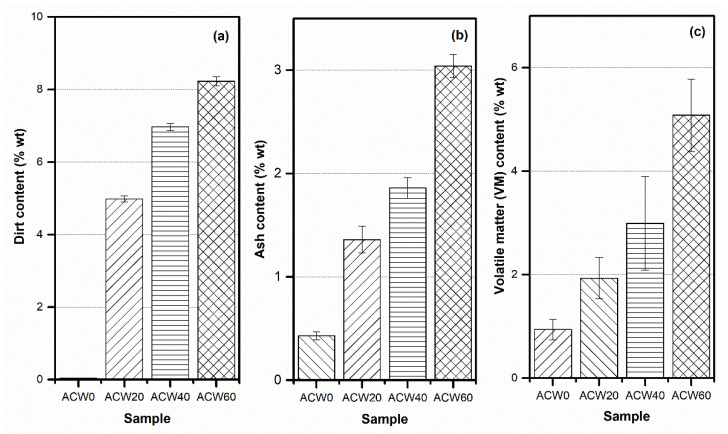
Dirt (**a**), ash (**b**) and volatile matter (VM) contents (**c**) of natural rubber as function Table 4. (i.e., 0, 20, 40 and 60 phr).

**Figure 4 polymers-13-04178-f004:**
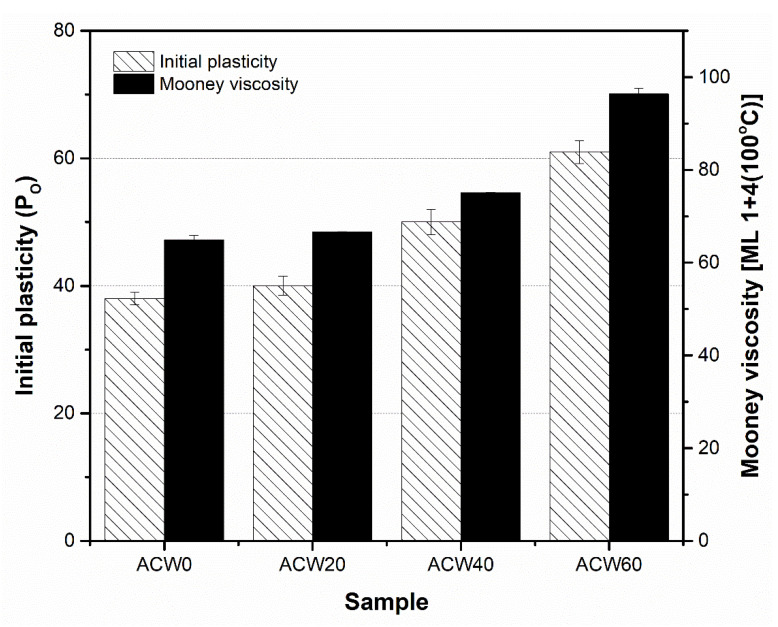
Initial plasticity and Mooney viscosity of natural rubber as a function of *Aquilaria crassna* wood content (i.e., 0, 20, 40, and 60 phr).

**Figure 5 polymers-13-04178-f005:**
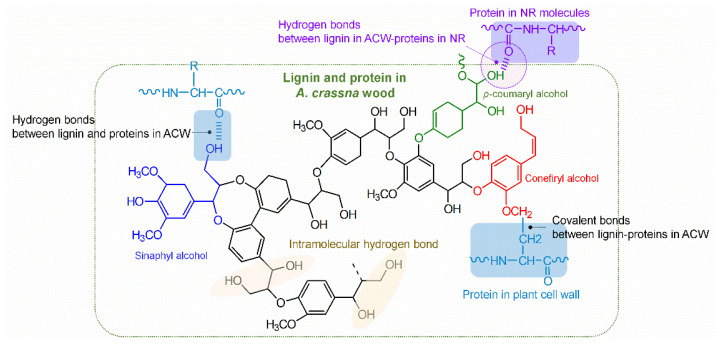
A proposed model for the interactions of lignin and plant proteins in the cell walls of *Aquilaria crassna* wood lignin with proteins in *Hevea* NR.

**Figure 6 polymers-13-04178-f006:**
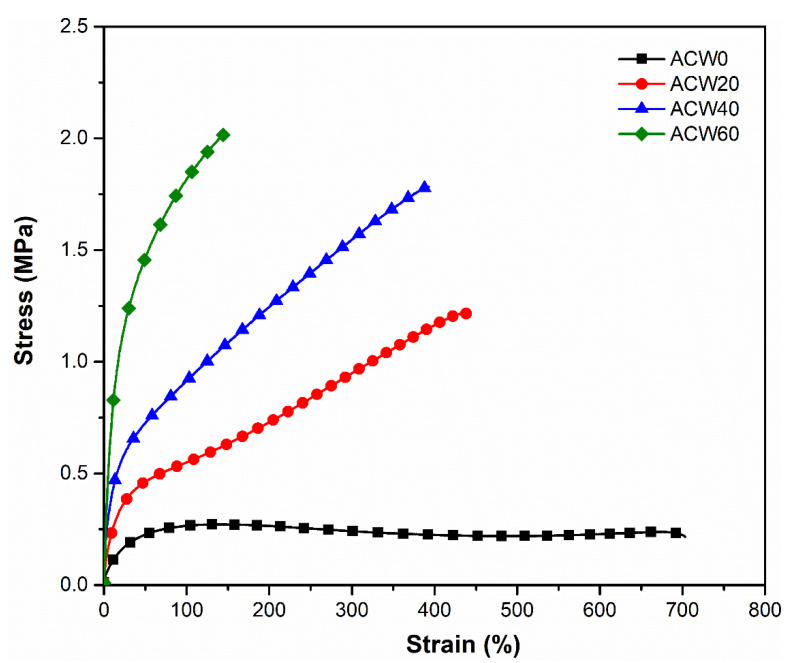
Stress–strain behavior of the ADS films as functions of *Aquilaria crassna* wood content (i.e., 0, 20, 40, and 60 phr).

**Figure 7 polymers-13-04178-f007:**
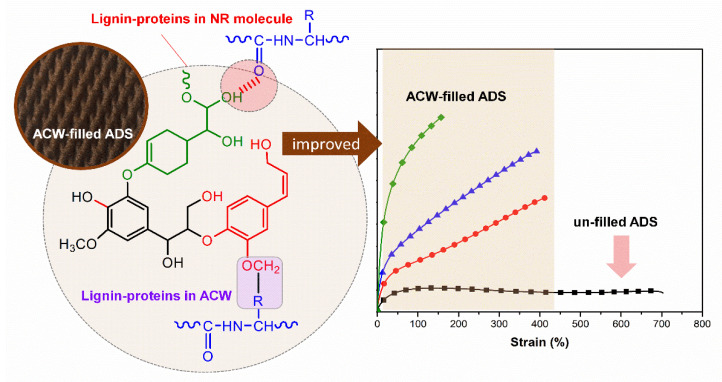
Possible physical and chemical interactions between lignin and proteins in NR.

**Figure 8 polymers-13-04178-f008:**
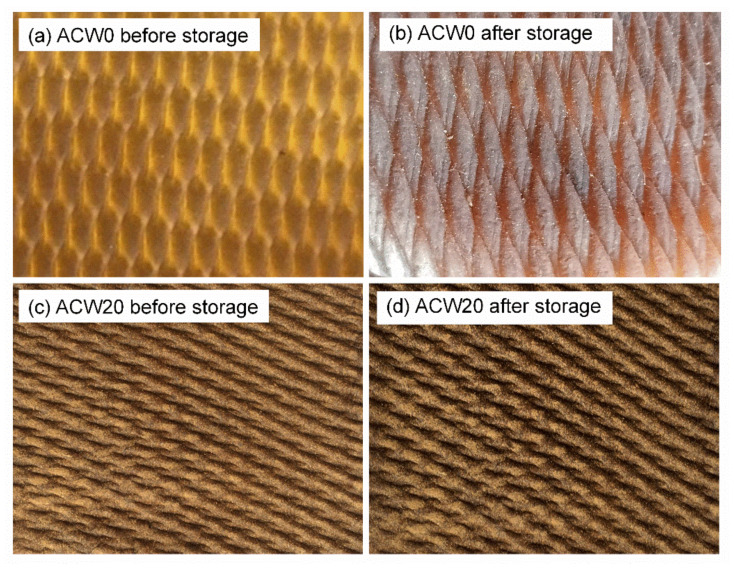
ACW0 and ACW20 filled ADS samples before and after storage (**a**–**d**) for 5 months at ambient temperature.

**Figure 9 polymers-13-04178-f009:**
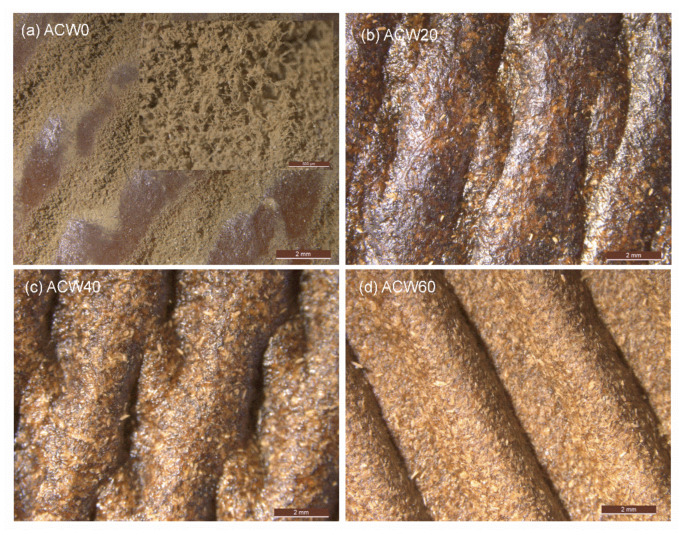
Morphological properties of the ACW0 (**a**), ACW20 (**b**), ACW40 (**c**), and ACW60 (**d**) samples after storage for 5 months at ambient temperature.

**Figure 10 polymers-13-04178-f010:**
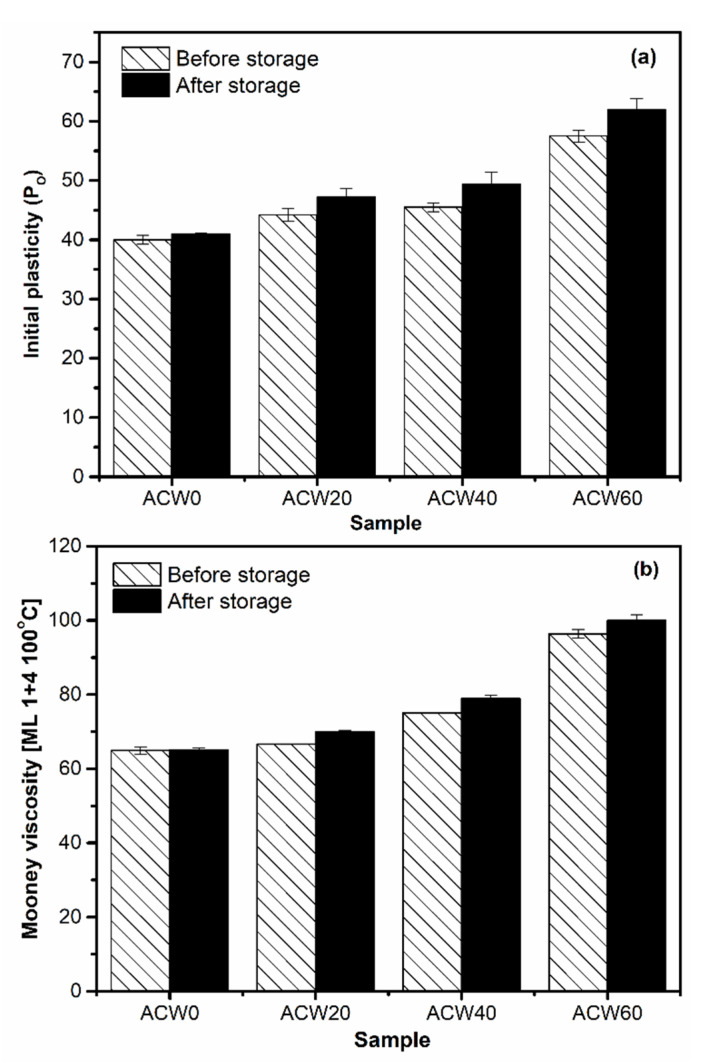
Initial plasticities (*P_O_*) (**a**) and Mooney viscosities (**b**) of the ADS samples as a function Figure 4. (i.e., 0, 20 40, and 60 phr) after storage for 5 months.

**Table 1 polymers-13-04178-t001:** Formulation of *Aquilaria crassna* wood dispersion.

Ingredient	Parts per Hundred of Rubber (phr)
*Aquilaria crassna* wood powder	20
bentonite	1
vultamol	1
water	78

**Table 2 polymers-13-04178-t002:** The FTIR band assignment of *Aquilaria. crassna* wood and natural rubber.

Wavelength (cm^−1^)	Assignment
1240	Amide III band (in-phase combination of C–N stretch and N–H bend modes) [21]
1308	CH_2_ wagging [19]
1375	CH_3_ asymmetric deformation [19]
1448	CH_2_ deformation [19]
1509	C–H stretching vibration of wood [13]
1540	Amide II: N–H and C–N stretching vibration of proteins in NR molecules [19]
1620	Aromatic ring stretching vibration [13]
1630	Amide I: R_1_–(C=O)–NH–R_2_ stretching vibration of proteins in NR molecules [19]
1660	C=C stretching vibration of *cis*-1,4-isoprene units [19]
1710	–(C=O)–OH stretching vibration of lipids in natural rubber [19]
1740–1720	C=O stretching vibration of aldehydes [13]
3280	N–H stretching vibration of proteins [19]

**Table 3 polymers-13-04178-t003:** Quantification of major elements in *Aquilaria crassna* wood.

Element	Concentration (%)
C	27.359
O	36.444
N	31.905
H	2.296
Na	0.034
Mg	0.076
Al	0.104
Si	0.195
P	0.043
S	0.145
Cl	0.136
K	0.243
Ca	0.874
Ti	0.011
Mn	0.028
Fe	0.090
Cu	0.011
Zn	0.006
Sr	0.002

**Table 4 polymers-13-04178-t004:** Physical properties of natural rubber with various *Aquilaria crassna* wood-filler contents (i.e., 0, 20, 40 and 60 phr).

Property\Sample	ACW0	ACW20	ACW40	ACW60
dirt content (% wt)	0.03 ± 0.06	6.88 ± 0.06	6.97 ± 0.06	8.23 ± 0.06
ash content (% wt)	0.43 ± 0.04	1.36 ± 20.04	1.86 ± 40.04	3.04 ± 60.04
volatile matter (VM) content (% wt)	0.94 ± 0.2	01.93 ± 0.40	2.99 ± 0.90	5.08 ± 1.00
nitrogen content (% wt)	0.410 ± 0.020	0.447 ± 0.122	0.487 ± 0.070	0.513 ± 0.115

**Table 5 polymers-13-04178-t005:** Initial plasticities (*P_O_*), plasticity retention indexes (PRI), and Mooney viscosities of the rubber sheets, along with their gel contents.

Sample	Initial Plasticity (*P_O_*)	Plasticity Retention Index (PRI)	Mooney Viscosity [ML 1+4 (100 °C)]	Gel Content (%wt)
ACW0	38.2 ± 0.7	100.6 ± 2.0	64.9 ± 1.0	8.78 ± 0.14
ACW20	40.1 ± 1.1	91.2 ± 3.0	66.6 ± 0.1	20.53 ± 0.75
ACW40	50.1 ± 0.8	64.0 ± 1.0	75.1 ± 0.1	31.57 ± 1.85
ACW60	61.2 ± 1.0	45.1 ± 1.5	96.4 ± 1.2	50.22 ± 1.19

**Table 6 polymers-13-04178-t006:** Mechanical properties of the NR films.

Property\Sample	ACW0	ACW20	ACW40	ACW60
100% modulus (MPa)	0.26 ± 0.01	0.56 ± 0.02	0.95 ± 0.01	1.94 ± 0.03
300% modulus (MPa)	0.24 ± 0.02	0.99 ± 0.04	1.61 ± 0.03	-
Green strength (MPa)	0.27 ± 0.01	1.29 ± 0.02	1.84 ± 0.01	2.19 ± 0.03
Elongation at break (%)	669.52 ± 29.17	434.22 ± 17.08	386.92 ± 11.16	158.36 ± 15.77

**Table 7 polymers-13-04178-t007:** Fungal growth area fractions at different *Aquilaria crassna* wood (ACW) contents in air-dried sheets (ADS).

*Aquilaria crassna* Wood Content (phr)	% Area of Fungal Growth
0	97.62 ± 2.45
20	73.70 ± 1.65
40	69.11 ± 0.56
60	65.23 ± 1.54

## Data Availability

The data presented in this study are available on request from the corresponding author.

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
