# Peer review of "Study of Aquilaria crassna Wood as an Antifungal Additive to Improve the Properties of Natural Rubber as Air-Dried Sheets"

_polymers, 2021, doi:10.3390/polym13234178_

Round 1
Reviewer 1 Report
The manuscript reported the aquilaria crassna wood used as additive to improve the antifungal, physical and mechanical properties of natural rubber as air dried sheets. The manuscript is clear and logical for reading. It is suggested to be published after a minor revision.
1. Considering the ACW powder was only passed through a 120-mesh screen, the reviewer wonder if the effect of particle size and particle size distribution of ACW on NR. After all, such a high particle size is not widely used in rubber composites.
2. Moreover, in Line 57, it is recommended to add the “2.1 Materials” in “2. Materials and Methods”.
3. The superscripts and subscript were also irregular; There should be a space between the number and unit.
Author Response
We would like to express our appreciation to the reviewers’ careful reading of the manuscript. We have found their comments to be very helpful in improving the quality of the manuscript. In the following, the reviewers’ comments are given in black typeface and our responses immediately follow in blue for reviewer. The revision parts in the revised manuscript were blue typeface.

Reviewer 2 Report
I read an interesting and comprehensive research work entitled Study of Aquilaria crassna Wood as Additive to Improve the Properties of Natural Rubber as Air Dried Sheets. The concept of the article is interesting and suitable to publish in Polymers. This manuscript is generally well written and clearly presented however still needs to address some comments, and thus require moderate revision to improve the quality of the manuscript.Title should modify which can describe whole research work mention which properties? Add potential word before additive
- Abstract looks very general authors should mention the importance of research work briefly. Add more discussion related to the research, values of research output and future research directions as well.
- Add more keywords.
- A well addressed graphical scheme of study design should be inserted.
- In the introduction section write the novelty of the work and the problem statement clearly. More discussion about the other techniques used to Improve the Properties of Natural Rubber are still needed. Give detailed research objectives at the end of introduction not the repetition of abstract.
- Statistical analysis of the results should be provided in the materials and methods section. It's important for all experimental work Report these values in the results and discussion.
- This manuscript lacking substantial discussion of results with the literature authors should concentrate on this during revision. Add more recent references of year 2018-2021.
- Techno Economic challenges of the developed system and future research directions that need to be described by adding a new section before the conclusions section?
- The conclusion of the study needs to be added with the specific output obtained from the study, it could be modified with precise outcomes with a take home message.
- Some English and grammar mistakes are present that need to be correct to improve the quality of the manuscript.
Author Response

(The authors gave the same response as above.)

Round 2
Reviewer 2 Report
The authors have substantially revised the manuscript according to the comments.
The present form of the manuscript can be accepted for publication.